# Humoral and cellular responses after a third dose of SARS-CoV-2 BNT162b2 vaccine in patients with lymphoid malignancies

Daniel Re [1,2,12✉], Barbara Seitz-Polski [3,4,5,12], Vesna Brglez [3,4,5], Michel Carles [6,7], Daisy Graça[3,4,5], Sylvia Benzaken[3,4,5], Stéphane Liguori[8], Khaled Zahreddine[8], Margaux Delforge[9], Béatrice Bailly-Maitre[7], Benjamin Verrière[9], Emmanuel Chamorey[10] & Jérôme Barrière [11✉]

Patients with hematological malignances have impaired immune response after two doses of BNT162b2 (Pfizer/BioNTech) vaccine against SARS-CoV-2. Here, in this observational study (registration number HDH F20210324145532), we measure SARS-CoV-2 anti-Spike antibodies, neutralizing antibodies and T-cell responses after immune stimulation with a third dose (D3) of the same vaccine in patients with chronic lymphocytic leukemia ($n = 13$), B cell non-Hodgkin lymphoma ($n = 14$), and multiple myeloma ($n = 16$)). No unexpected novel side effects are reported. Among 25 patients with positive anti-S titers before D3, 23 (92%) patients increase their anti-S and neutralizing antibody titer after D3. All 18 (42%) initially seronegative patients remain negative. D3 increases the median IFN-γ secretion in the whole cohort and induces IFN-γ secretion in a fraction of seronegative patients. Our data thus support the use of a third vaccine dose amongst patients with lymphoid malignancies, even though some of them will still have vaccine failure.

[1] Department of Medical Oncology, Centre Hospitalier, Antibes, France. [2] Department of Medical Oncology, Centre Antoine Lacassagne, Nice, France. [3] Laboratoire d'Immunologie, Centre Hospitalier Universitaire (CHU) de Nice, Université Côte d'Azur, Nice, France. [4] Unité de Recherche Clinique de la Côte d'Azur (UR2CA), Université Côte d'Azur, Nice, France. [5] Centre de Référence Maladies Rares Syndrome Néphrotique Idiopathique, CHU de Nice, Université Côte d'Azur, Nice, France. [6] Department of Infectious Disease, Centre Hospitalier Universitaire de Nice (CHU), Nice, France. [7] INSERM, C3M, Université Côte d'Azur, Nice, France. [8] Department of Medical Biology, Centre Hospitalier, Antibes, France. [9] Department of Pharmacy, Centre Hospitalier, Antibes, France. [10] Department of Biostatistics and Epidemiology, Centre Antoine Lacassagne, Nice, France. [11] Department of Medical Oncology, Polyclinique Saint-Jean, Cagnes-sur-Mer, France. [12] These authors contributed equally: Daniel Re, Barbara Seitz-Polski. ✉email: daniel.re@ch-antibes.fr; j.barriere@polesantesaintjean.fr

Patients suffering from solid cancer (SC) or hematological malignancies (HM) are at increased risk of severe Coronavirus disease (COVID-19)[1,2], caused by an infection with severe acute respiratory syndrome coronavirus 2 (SARS-CoV-2). HM include mainly myeloid and lymphoid diseases that respond differently to infection depending on the cell of origin of the underlying disease and the current or past treatment used to control HM. Some of these treatments such as monoclonal B-cell depleting anti-CD20 antibodies are known to alter patient's immune response and sometimes durably. This is highlighted by the observation that vaccination against SARS-CoV-2 is less effective in immunocompromised patients such as organ transplanted patients[3], patients treated for SC[4–6] or patients with HM and especially patients with chronic lymphocytic leukemia (CLL)[7–9] in comparison to a healthy population. Patients with lymphoid malignancies need thus a special attention during COVID-19 pandemic.

Others and our team previously showed that patients treated with anti-CD20 monoclonal antibodies (Mabs) prior to COVID-19 vaccination had a very low likelihood of developing a humoral response, especially if the anti-CD20 Mab treatment is administrated within 12 months before the administration of the anti-SARS-CoV-2 vaccine[8,9]. T-cell response seem more impaired in patients with HM than with SC, with a beneficial effect of the booster dose[6].

In view of the altered immune response of patients suffering from HM we decide to conduct a specific observatory of patients followed at our hospital for lymphoid malignancies (LM) after vaccination with two doses of the SARS-CoV-2 mRNA BNT162b2 vaccine (Pfizer / BioNtech).

Here, we report the monitoring of humoral and cellular responses to a third vaccine dose (dose 3), in patients with poor or no response to two previous vaccine doses. We show that a third booster dose increases antibody titers and neutralizing antibodies in patients with multiple myeloma (MM) but less likely in CLL or recently anti-CD-20-treated patients with indolent and aggressive B-cell non-Hodgkin lymphoma (NHL). Despite the absence of seroconversion in our population of selected patients with CLL and NHL, we are able to demonstrate the presence of a possibly protective cellular T-cell response in these fragile patients. The possibility to boost cellular responses in patients without antibody response has to be considered when proposing novel vaccine strategies to immunocompromised patients.

## Results

**Patient characteristics**. We analyzed a data set of 45 patients, prospectively included to receive dose 3 of the BNT162b2 vaccine given 78 days [range: 47–114] after dose 2 of the same vaccine. Included patients were suffering from CLL ($n = 15$), NHL ($n = 14$), and MM ($n = 16$). All 45 patients were negative for anti-N Abs before dose 3, but two patients with CLL were tested positive after dose 3, suggesting a virus-related immune stimulation and therefore excluded from the final analyses. The median age of the 43 remaining patients was 77 years [range: 37–92], 63% were men and 37% women (Table 1). Concomitant treatment of patients at the moment of administration of dose 3 are reported in Table 1. Assays used to analyze humoral and cellular response in the respective patient cohorts are detailed in Fig. 1.

*Humoral immunity*. Among 43 patients, 18 (41.8%) had no total anti-S Abs before dose 3 of the BNT162b2 vaccine ($n = 9/13$ (69%) patients with CLL, $n = 8/14$ (57%) patients with NHL, $n = 1/16$ (6%) patients with MM), and all 18 remained negative after the dose 3 (Table 1). Fourteen of these 18 patients had

already received an anti-CD20 Mab treatment, nine of them within the 12 months before the vaccination. One seronegative patient with MM was under active treatment for human immunodeficiency virus (HIV) infection.

In univariate analysis, age and type of LM, but not sex or type of treatment (except for anti-CD20 Mab within 12 months before the administration of the anti-SARS-CoV-2 vaccine), were statistically associated with anti-S Abs response after dose 3 (Table 1).

Among the 25 patients (58.1%) with positive anti-S titers before dose 3 ($n = 4$ CLL, $n = 6$ NHL, $n = 15$ MM), all patients remained positive (100%) and 23 patients (92%) increased their median anti-S titer after dose 3 from 87.1 U/mL [range: 1.2–693] to 3386 U/mL [range: 6.6–20312] ($p < 0.001$) (Table 1). Figure 2A shows the median anti-S titer after dose 2 and dose 3 according to the respective pathology: 0 U/mL [range: 0–120] and 0 U/mL [range: 0–5997] ($p = 0.12$) in patients with CLL, 0 [range: 0–310] and 0 U/mL [range: 0–6101] ($p = 0.07$) in patients with NHL, 100 U/mL [range: 0–690] and 2700 U/mL [range: 0–20,312] ($p < 0.0001$) in patients with MM. We then focused on 23 patients with a history of anti-CD20 treatment: patients treated within 12 months before the administration of dose 3 responded poorly (median anti-S titer: 0 U/mL [range: 0–6101]) when compared to patients receiving the same drug at least 12 months before the BNT162b2 vaccine (median anti-S titer: 4200 [range: 0–6073]) ($p = 0.047$) (Fig. 2B).

A surrogate virus neutralization assay was performed to analyze the capacity of patients' anti-S Abs to block the entry of SARS-CoV-2 into the cells by blocking its binding to ACE-2 receptor. When looking at the whole patient population ($n = 43$), we globally found a very good correlation between total anti-S titers shown in Fig. 2 and the inhibition capacity of patients' neutralizing Abs before (Fig. 3A) and after dose 3 (Fig. 3B) ($p < 0.0001$ for both timepoints). Among 25 patients with measurable total anti-S titers before and after dose 3, 11 (44%) had neutralizing Abs before dose 3. This number increased to 21 (84%) after dose 3 (Fig. 3C, $p = 0071$). The neutralizing capacity of Abs in positive patients was boosted from 18.5% (range [0.0–92.7]) to 96.3% (range [0.0–98.3]); Fig. 3D). However, four patients with detectable anti-S titers were inferior to the positive threshold for neutralizing Abs. Total anti-S titer after dose 3 was lower in these four patients (72.4 U/mL, range [6.6–156]) when compared to the 21 positive patients (3690 U/mL, range [465–20,312]; $p = 0.0002$). In line with these findings, 18 patients with no detectable anti-S Abs before and after dose 3 were also negative for neutralizing Abs at both timepoints (Fig. 3A, B).

**Exploratory comparative cellular immunity response with humoral response**. We next aimed to characterize specific T-cell responses to the BNT162b2 vaccine with a whole-blood IGRA test in all 27 patients with CD-20-positive disease to assess comparative immune response among poor humoral immune responders. We were able to collect and analyze blood samples in 22 patients (CLL $n = 10$, NHL $n = 12$). Anti-S response of this subgroup to dose 3 is shown in Supplementary Fig. 1.

Among nine patients with positive anti-S Abs, six showed a T-cell response before dose 3 (66.6%) and administration of dose 3 did not change this number (Fig. 4). In contrast, only five out of 13 patients (38.5%) without anti-S Abs had a positive Quantiferon assay before dose 3. The number of patients with a positive Quantiferon assay increase to eight (61.5%) after the booster dose leaving a double-negative population of five out of 22 patients (22.7%) without neither a T-cell response nor measurable anti-S Abs. These five patients included three cases

**Table 1 Patient characteristics of patients included in this study, before and after dose 3 (same data).**

| SARS-CoV-2 anti-S antibody response | Negative < 0.8 U/mL | Positive ≥ 0.8 U/mL | p |
|---|---|---|---|
| Number of patients analyzed | 18 | 25 | |
| Median anti-S Abs titer before dose 3 (range) | 0 (0,0) | 87.1 (1.2–693) | |
| Median anti-S Abs titer after dose 3 (range) | 0 (0,0) | 3386 (6.6–20,312) | |
| Age | | | 0.031 |
| <70 | 2 (18.2%) | 9 (81.8%) | |
| ≥70 | 16 (50%) | 16 (50%) | |
| Sex | | | 1 |
| Male | 11 (40.7%) | 16 (59.3%) | |
| Female | 7 (43.8%) | 9 (56.2%) | |
| Type of lymphoid malignancies | | | 0.001 |
| CLL | 9 (69.2%) | 4 (30.8%) | |
| NHL | 8 (57.1%) | 6 (42.9%) | |
| MM | 1 (6.2%) | 15 (93.8%) | |
| Last type of treatment | | | 0.19 |
| Anti-CD20 Mab + chemotherapy | 9 (56.2%) | 7 (43.8%) | |
| Anti-CD20 Mab + Venetoclax | 3 (100%) | 0 (0%) | |
| Tafasitamab + Lenalidomide | 0 (0%) | 1 (100%) | |
| Chemotherapy | 1 (100%) | 0 (0%) | |
| Venetoclax | 1 (100%) | 0 (0%) | |
| Imbruvica | 3 (60%) | 2 (40%) | |
| Anti-CD38 antibody combination | 0 (0%) | 7 (100%) | |
| IMID | 1 (25%) | 3 (75%) | |
| Ixazomib | 0 (0%) | 1 (100%) | |
| Never treated | 0 (0%) | 4 (100%) | |

*CLL* chronic lymphocytic leukemia, *NHL* indolent and aggressive B-cell non-Hodgkin lymphoma, MM multiple myeloma, Mab monoclonal antibody, IMID immune modulatory drug.

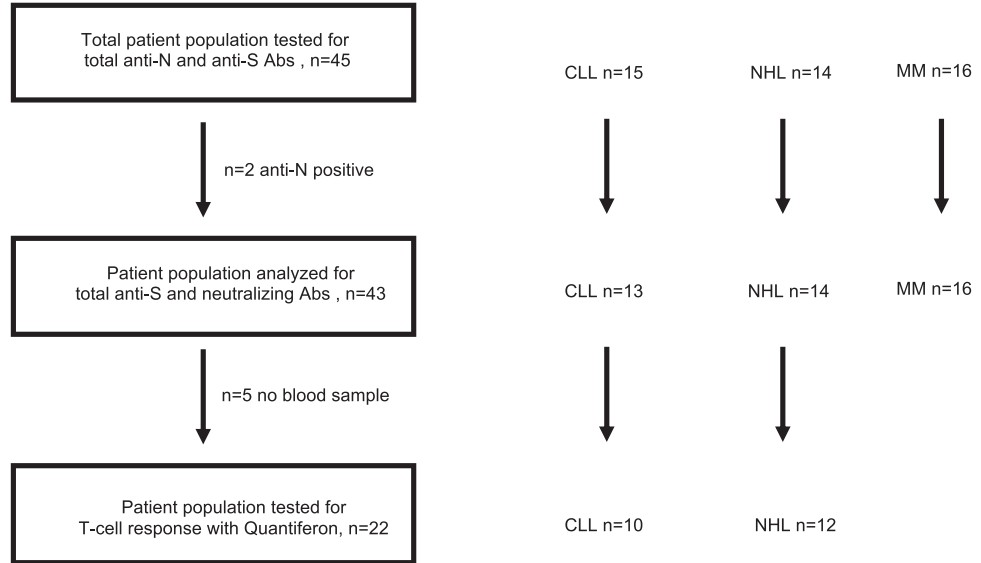

**Fig. 1 Flowchart of patient inclusion and exclusion in the study.** CLL chronic lymphocytic leukemia, NHL B-cell non-Hodgkin lymphoma, MM multiple myeloma, anti-S Abs total anti-SARS-CoV-2 Spike antibodies, anti-N Abs total anti-SARS-CoV-2 Nucleocapsid antibodies.

of CLL (all actively treated with Venetoclax and a past history of Rituximab treatment) and two cases of NHL (one case untreated and one case treated with Rituximab and Bendamustine). Four of these five double-negative patients were under active treatment.

Dose 3 of the BNT162b2 vaccine increased median IFN-γ secretion after exposition to antigen 1 or antigen 2 from 0.07 IU/mL [range: 0.0–0.17] to 0.3 IU/mL [range: 0.0–0.9] ($p = 0.0008$) and from 0.06 IU/mL [range: 0.0–0.1] to 0.2 IU/mL [range: 0.0–1.3] ($p = 0.0006$), respectively (Fig. 5A). The timing of treatment with an anti-CD20 Mab impacted the humoral response as reported before, it did not modify the T-cell response: median of IFN-γ secretion after exposition to antigen 1 or antigen 2 was 0.14 IU/mL [range: 0.0–3.1] and 0.09 IU/mL [range: 0.0–0.9], respectively, for patients treated at least 12 months before the vaccine administration, versus 0.5 IU/mL [range: 0.0–1.1] and 1.2 IU/mL [range: 0.0–3.3], respectively, for patients treated within 12 months prior to vaccine administration ($p = 0.459$ and $p = 0.479$, respectively) (Fig. 5B). Patients on active NHL or CLL treatment during the vaccination sequence had a poorer specific T-cell response than patients without ongoing cancer specific medication. The median of IFN-γ secretion after exposition to antigen 2 was 0.0 IU/mL [range: 0.0–0.5] vs. 0.9 IU/mL [range: 0.1–4.0] $p = 0.049$ ($p = 0.08$ for

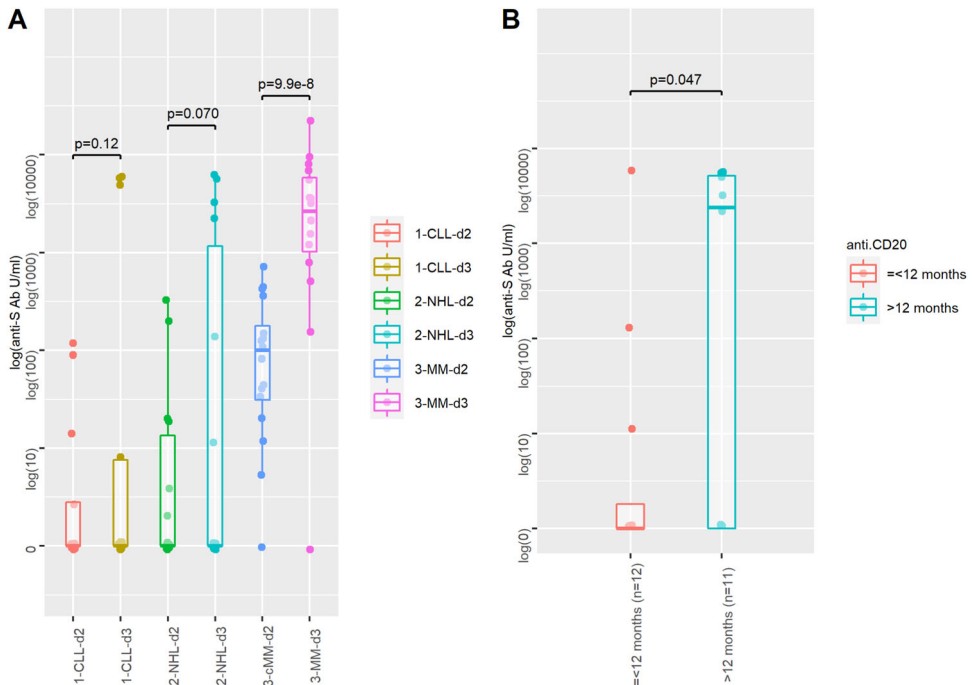

**Fig. 2 Humoral quantitative anti-Spike (S) antibodies (logarithmic scale) response to the BNT162b2 vaccine. A** Reponse after dose 2 (d2) and after dose 3 (d3) in 43 patients with lymphoid malignancies (n = 13 patients with chronic lymphocytic leukemia (CLL), n = 14 patients with B-cell non-Hodgkin lymphoma (NHL) and n = 16 patients with multiple myeloma (MM)). **B** Response after dose 3 in 23 patients pre-treated with an anti-CD20 Mab within 12 months prior to vaccine administration (n = 12) or at least 12 months prior to vaccine administration (n = 11). The upper whisker extends from the hinge to the highest value that is within 1.5 * IQR of the hinge, where IQR is the inter-quartile range, or distance between the first and third quartiles. The lower whisker extends from the hinge to the lowest value within 1.5 * IQR of the hinge. Data beyond the end of the whiskers are outliers and plotted as points (as specified by Tukey). p-value (two-sided): paired t-test after data normalization using logarithm. No adjustment was done for multiple comparison. Source data are provided as a Source Data file.

antigen 1) (Fig. 5C). There was no difference in T-cell response between patients with CLL or NHL (p > 0.99) and their respective T-cell response after dose 3 is shown in Supplementary Fig. 2.

We then compared the performance of serologic and IGRA testing using the method of the receiver operating characteristic (ROC) curve to explore the best way to detect a specific immune response to SARS-CoV-2 vaccine. In this cohort of 22 patients (and seven healthy subjects naive for SARS-Cov2 infection as negative controls), the IGRA test based on two different antigens identified more efficiently than the Elecsys ® Anti-SARS-CoV-2 immunoassay a specific immune response to SARS-CoV-2 vaccine (aera under the curve (AUC) anti-S Abs: 0.7045, p = 0.11; AUC IGRA Antigen 1: 0.8636, p = 0.004 (Sensitivity: 65%, Specificity: 100%); AUC IGRA Antigen 2: 0.8864, p = 0.002, (Sensitivity: 60%, Specificity: 100%)).

**Immune response after two or three doses compared to healthy donors after two doses of BNT162b2 vaccine.** Humoral and cellular response of CLL and NHL patients (n = 22) after two or three doses of BNT162b2 vaccine was compared to humoral and cellular response of healthy donors' controls (HD) after two doses of BNT162b2 vaccine awaiting the third dose (Fig. 6). While humoral (i.e., titer of anti-S Ab and rate of neutralization) and cellular response after two doses of our patients were weak compared to HD (Titer of anti-S Ab: 0.0 [0.0; 16.6] vs. 548.5 [255.0; 1364.0] p < 0.0001, Fig. 6A, Neutralization: 14% [0; 85] vs. 46 [7: 72] p = 0.08, Fig. 6B) (IGRA Antigen 1: 0.07 [0.0; 0.2] vs. 0.2 [0.1; 0.3] p = 0.02, Fig. 6C and IGRA Antigen 2: 0.05 [0.0; 0.1] vs. 0.4 [0.13;1.0] p = 0.009, Fig. 6D), dose 3 tended to restore a comparable cellular response to HD treated by two

doses (Titer of anti-S Ab: 0.0 [0.0; 2626] vs. 548.5 [255.0; 1364.0] p = 0.2, Fig. 6A, Neutralization: 91% [17; 99] vs. 46 [7: 72] p = 0.09, Fig. 6B) (IGRA Antigen 1: 0.3 [0.0; 0.9] vs. 0.2 [0.1; 0.3] p = 0.9, Fig. 6C and IGRA Antigen 2: 0.2 [0.0; 1.3] vs. 0.4 [0.13;1.0] p = 0.46, Fig. 6D; Supplementary Fig. 3: results confirmed with age matching).

**Tolerance of the dose 3.** No additional adverse events were noticed in our population, during the 3 to 5 weeks of follow-up after dose 3 of BNT162b2 vaccine. No adverse events of grade 3–4 were reported with only grade 1–2 transient effects, all resolved at follow-up visit.

**Discussion**

To our knowledge, this study is the first to report results on cellular and humoral immunity after administration of a third dose of BNT162b2 vaccine in patients treated for LM. To date, only three studies have shown a favorable impact of the administration of a third dose of BNT162b2 vaccine to solid-organ transplant recipients with a significantly improved humoral response[10–12].

Previous reports highlighted reduced rates of seroconversion after two doses of a SARS-CoV-2 vaccine for patients with HM, leaving some of them without detectable anti-S Abs response[7–9], with lower titers for patients with CLL, even without treatment, or patients under anti-CD20 therapy. We here show that a third dose does not induce seroconversion in patients who have not responded before to two doses of the same vaccine. Therefore, anti-S level measurement before and after a third vaccination seems useless in current clinical practice for this specific population. Conversely, the third vaccine dose increased the overall

**Fig. 3 Correlation between the anti-S Abs titer (logarithmic scale) and the neutralization of S1/RBD binding to ACE-2 receptor. A** Correlation in 43 patients with lymphoid malignancies before (spearman test ($i = 0.88$, $p < 0.0001$) and **B** 3 to 5 weeks after the administration of the BNT162b2 vaccine (dose 2 and 3, respectively) (spearman test ($r = 0.94$, $p < 0.0001$). Patients without a detectable Abs titer were assigned a value 0.1 U/mL. **C** Proportion of patients with detectable anti-S Abs titers after dose 2 or dose 3 that inhibited the S1/RBD binding to ACE-2 receptor (Fisher's exact test, two-sided ($p < 0.0072$)). Patients' sera with at least 20% inhibition of binding were considered as positive. **D** Evolution of neutralization capacity of 25 patients with detectable anti-S Abs titers after dose 2 or dose 3 (Wilcoxon matched-pairs signed rank test ($p < 0.0001$)). Source data are provided as a Source Data file.

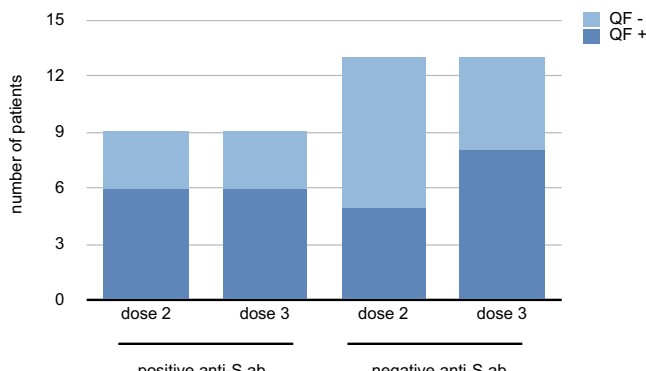

**Fig. 4 Number of quantiferon (QF)-positive patients before and after a third dose of the BNT162b2 vaccine.** It is shown the number of chronic lymphocytic leukemia and B-cell non-Hodgkin lymphoma patients ($n = 22$) with or without total anti-SARS-CoV-2 Spike antibodies (anti-S Abs) depending on the number of doses of the vaccine administered. Source data are provided as a Source Data file.

humoral response in patients who had detectable Abs after the second dose, especially for patients with MM, and to a lower extent for patients with NHL, but not for patients with CLL. These results suggest various level of specific B-cell responses, related to different HM. Further studies on larger cohorts are needed to confirm these results.

Our data shows a clear correlation of anti-S Abs with neutralizing Abs. Dose 3 both significantly increased the number of patients with neutralizing Abs from 44% to 84% and boosted the neutralizing capacity of the Abs response above 95% in seropositive patients, suggesting that vaccine-induced humoral response can confer a better degree of protection against Sars-CoV-2 infection, potentially similar to HD donors having received two vaccine doses.

Similar to patients with anti-CD20-treated multiple sclerosis[13], some of our patients with LM did not develop a specific T-cell response after the second dose. Conversely, several of the seronegative patients showed an emerging cellular response after dose 3, suggesting a stimulating effect of the second booster dose on the cellular response, despite the absence of humoral response. Given the importance of a T-cell response in critically ill COVID-19

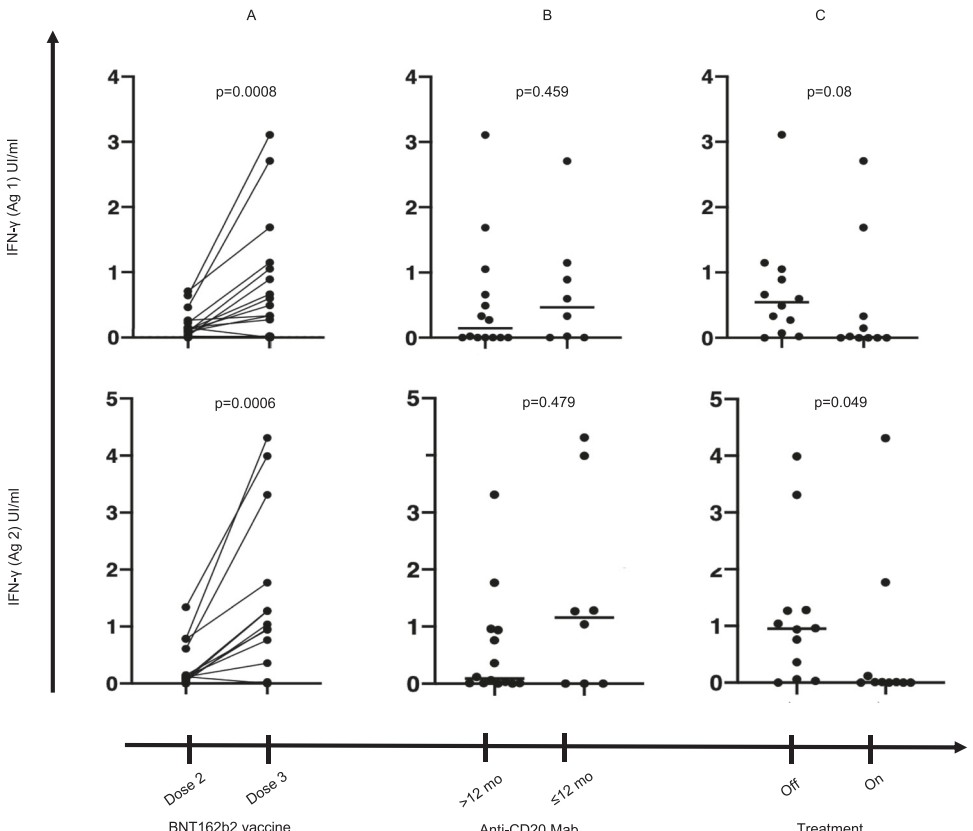

**Fig. 5 Level of IFN-gamma secretion after exposition to antigen 1 or antigen 2 of SARS-CoV-2.** It is shown the number of UI/mL of IFN-γ secretion in chronic lymphocytic leukemia and B-cell non-Hodgkin lymphoma patients (n = 22) **A** after a third dose of the BNT162b2 vaccine (Wilcoxon matched-pairs signed rank test: p = 0.0008 (antigen 1) and p = 0.0006 (antigen 2), **B** depending on the timing of anti-CD20 monoclonal antibody exposition (Mann–Whitney test: p = 0.459 (antigen 1) and p = 479 (antigen 2), **C** depending on concomitant administration of active lymphoma treatment (Mann–Whitney test: p = 0.08 (antigen 1) and p = 0.0487 (antigen 2)). Source data are provided as a Source Data file.

patients[14,15], and especially in patients followed for HM[16], this is another suggested benefit of the third vaccine dose. In particular, dose 3 may be highly beneficial for patients treated with anti-CD20 Mab-based therapy, already known to be at higher risk of death or prolonged SARS-CoV-2 shedding[17,18].

Our study has some limitations, mainly due to the small number of included patients. Data on cellular immunity have been restricted to a subgroup of the cohort due to availability issues, therefore precluding more comprehensive analyses of various cellular responses related to specific treatments.

Considering the decrease of humoral immunity over time[19], the known correlation between neutralizing Abs titers and clinical response[20] and protection against Beta and Delta variants[21], our results showing the rise of a specific cellular T-cell response, an early third dose of the SARS-CoV-2 vaccine appears to be beneficial in patients with LM to improve anti-viral immunity. Patients treated with anti-CD20 or with drugs that are toxic to stem cells such as Bendamustine, remain of concern since they demonstrated a lesser humoral response to dose 3.

The fact that we did not observe seroconversion after dose 3 in patients without seroconversion after dose 2 will require additional confirmatory data. This does not mean that these patients should not be vaccinated, since dose 3 allowed stimulation of a T response, at least in some of them. For these patients, in addition to the incentive of relatives to get vaccinated and the drastic maintenance of social protection measures, repeated immune stimulation with a fourth vaccine dose, a multimodal immune stimulation with heterologous prime-boost vaccination, or a

maximized immune stimulation double-dose approach should be considered[22].

## Methods

**Patient population.** All eligible patients already participated in a previous observational study[9]. A third dose of the BNT162b2 vaccine was administered on a voluntary basis to patients with HM receiving active lymphodepleting treatment or being at risk to require further treatment. A cohort of 10 healthy donors' controls (HD) was also constituted (matched on sex ratio, median age 64 years old, treated by two doses of the BNT162b2 vaccine before the third dose as recommended by the French Authorities). All participants signed a written informed consent and accepted their participation in this registered observatory in accordance with ethical and legal French policies (Health Data Hub Registration number F20210324145532, https://www.health-data-hub.fr/projets/suivi-serologique-post-vaccination-sars-cov-2) that has been approved by our local ethics committee (« Comité d'Éthique du Centre Hospitalier Antibes-Juan les Pins »). The study design and conduct complied with all relevant regulations regarding the use of human study participants and was conducted in accordance with the criteria set by the Declaration of Helsinki. We excluded patients with a history of allergic reaction to the BNT162b2 vaccine or to polyethylene glycol. Included patients were vaccinated with the third dose in May 2021 at the hospital ambulatory oncology service. We followed standard operating procedures for vaccine administration and blood sampling in accordance with recommendations of the French Authorities published in April 2021[23]. All data were prospectively collected on an electronic case report form. Each included patient was informed of potential vaccine side effects and encouraged to report them. Tolerability was assessed 3 to 5 weeks after dose 3 by a medical interview and a physical examination during a follow-up visit.

**Humoral antibody responses.** Humoral responses were measured with Elecsys Anti-SARS-CoV-2 immunoassay (Roche Diagnostics, France) with detection of total antibodies (Abs) (immunoglobulin (Ig) G, IgA, IgM) against the SARS-CoV-2 Nucleocapsid (N) antigen (qualitative detection) and total Abs against the SARS-CoV-2 Spike (S) protein receptor binding domain (quantitative detection). For the anti-N assay, a serum index cutoff ≥ 1.0 was considered to be reactive and suggested a potential virus contact. For the anti-S assay, a serum index cutoff ≥0.8 U/

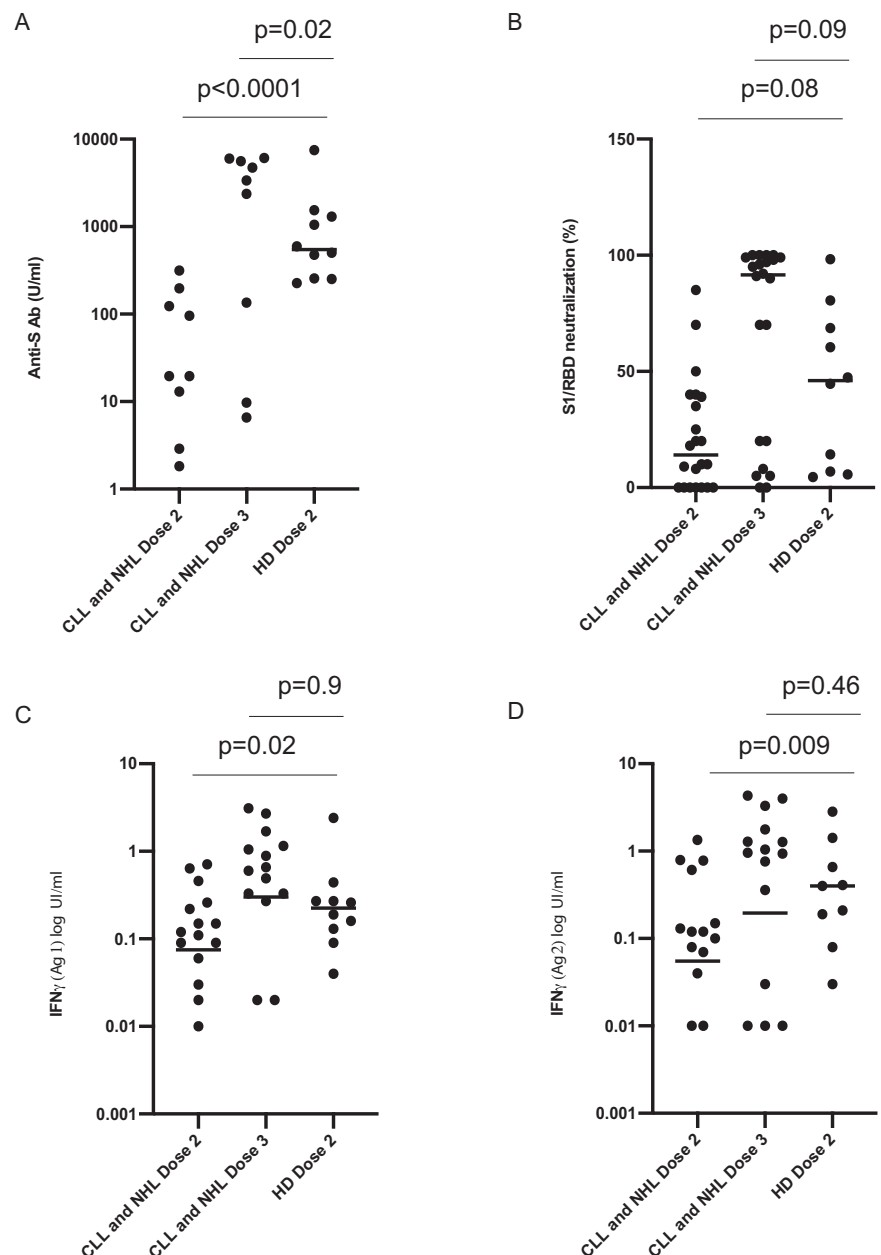

**Fig. 6 Immune responses in chronic lymphocytic leukemia and B-cell non-Hodgkin lymphoma (*n* = 22) and healthy donors (*n* = 10) after two and three doses of the BNT162b2 vaccine. A** Titer of anti-S Abs ($p < 0.0001$ using Mann–Whitney test); **B** Percentage of neutralization of S1/RBD binding to ACE-2 receptor (Mann–Whitney test: dose 2 vs. control $p = 0.08$ and dose 3 vs. control $p = 0.09$); **C** IFN-γ secretion after exposition to antigen 1 of SARS-CoV-2 (logarithmic scale) (Mann–Whitney test: dose 2 vs. control ($p = 0.02$) and dose 3 vs. control ($p = 0.9$); **D** IFN-γ secretion after exposition to antigen 2 of SARS-CoV-2 (logarithmic scale) (Mann–Whitney test: dose 2 vs. control (0.009) and dose 3 vs. control ($p = 0.46$)). Source data are provided as a Source Data file.

mL was retained positive and suggested an effective virus- or vaccine-related immune response. These assays consisted in a double-antigen sandwich electro-chemiluminescence and were performed on a Cobas e 601 automate (Roche). The level of SARS-CoV-2 anti-N and anti-S Abs was measured the day of administration of the dose 3 of the mRNA vaccine BNT162b2 and again three to 5 weeks later (median 27 days (d), range [21; 35]).

The neutralization capacity of patients' sera was measured using a surrogate virus neutralization assay NeutraLISA (Euroimmun) following the manufacturer's instructions[24,25]. Briefly; before and after the administration of the third vaccine dose, patients' sera were simultaneously incubated with the recombinant receptor binding domain (RBD) of the S1 subunit of Spike protein and the angiotensin converting enzyme-2 (ACE-2) receptor. The inhibition of S1/RBD binding to ACE-2 by patients' Abs was quantified as the reduction of signal in an enzyme-linked immunosorbent assay in comparison to the control sample without anti-SARS-

CoV-2 antibodies. Patients' sera with at least 20% neutralization capacity were considered as positive as recommended by the manufacturer.

**Cellular T-cell responses**. Patients with CD20-positive disease (known to poorly respond to two doses of the BNT162b2 vaccine) who accepted to be tested for cellular T-cell response had additional blood samples taken at these two timepoints. SARS-CoV-2-specific T-cell responses were assessed by a whole-blood interferon-γ (IFN-γ) Release Immuno Assay (IGRA) using two Qiagen proprietary mixes of SARS-CoV-2 S protein (antigen 1 and antigen 2) selected to activate both CD4 and CD8 T-cells. Briefly, venous blood samples were collected directly into the Quantiferon tubes containing either spike peptides or positive and negative controls. Whole blood was incubated at 37 °C for 16–24 h and centrifuged to separate plasma. Interferon gamma was measured in these plasma samples using enzyme-

linked immunosorbent assay tests (QuantiFERON Human IFN-γ SARS-CoV-2, Qiagen)[26]. The cutoff point for the ELISA was >0.13 International Unit (IU) IFN-γ/mL for antigen 1 and >0.12 IU IFN-γ/mL for antigen 2.

**Statistical analyses**. Continuous data are presented as median and range. The d'Agostino & Pearson normality test was used to determine if a variable followed or not a Gaussian Distribution. Continuous values were normalized using logarithm function when appropriate. Continuous values were normalized using logarithm function and were compared by Student $t$-test or Mann–Whitney test when appropriate. Spearman's correlation test was also used to measure the degree of association between two continuous variables. Categorical data were summarized using counts and percentages; they were compared by using $Chi^2$ test or Fisher-exact test when appropriate. Statistical analyses were performed using R.4.0.3 and GraphPad Prism 7.0 (GraphPad Software, Inc., San Diego, CA). All comparisons were two-tailed, and the differences were considered significant when $p$-value < 0.05.

**Reporting summary**. Further information on research design is available in the Nature Research Reporting Summary linked to this article.

## Data availability

All data are included in the Supplemental Information or available from the authors upon reasonable requests, as are unique reagents used in this Article. The raw numbers for charts and graphs are available in the Source Data file whenever possible. Source data are provided with this paper.

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

## Acknowledgements

This research was supported by a grant to BSP from Conseil Départemental des Alpes-Maritimes and from the Agence Nationale pour la Recherche AO-Flash-COVID. The funders had no role in the design and conduct of the study; collection, management, analysis, and interpretation of the data; preparation, review, or approval of the manuscript; and decision to submit the manuscript for publication. We acknowledge all nurses of the day clinic of the Antibes Hospital (Carmen, Elodie, Laurence, Nicole, Vinaj, and Valérie) and more specifically Sylvie Andreo. We last acknowledge patients for their participation. The analyses described are the responsibility of the authors and do not necessarily reflect the views or policies of the US Department of Health and Human Services. The mention of trade names, commercial products, or organisations does not imply endorsement by the US government.

## Author contributions

D.R., B.S.P., E.C., and J.B. had full access to all of the data in the study and take responsibility for the integrity of the data and the accuracy of the data analysis. Concept and design: All authors. Acquisition, analysis, or interpretation of data: D.R., B.S.P., V.B., M.C., D.G., S.B., S.L., K.Z., M.D., B.B.M., B.V., E.C., J.B. Drafting of the manuscript: D.R., B.S.P., M.C., E.C., and J.B. Critical revision of the manuscript for important intellectual content: D.R., B.S.P., V.B., M.C., D.G., S.B., S.L., K.Z., M.D., B.B.M, B.V., E.C., J.B. Statistical analysis: B.S.P., V.B., E.C. Administrative, technical, or material support: D.R, B.S.P., V.B., M.C., D.G., S.B., S.L., K.Z., M.D., B.B.M., B.V., E.C., J.B. Supervision: D.R., B.S.P., and J.B.

## Competing interests

The authors declare no competing interests.
