## [Peer Review File · Nature Communications]

Editorial Note: This manuscript has been previously reviewed at another journal that is not operating a transparent peer review scheme. This document only contains reviewer comments and rebuttal letters for versions considered at Nature Communications

REVIEWERS' COMMENTS

Reviewer #4 (Remarks to the Author):

The authors have addressed the reviewer concerns. The work is timely and will be of interest to the general community.

Reviewer #5 (Remarks to the Author):

This revised study attempts to assess the humoral and cell-mediated responses of different cancer patients (NHL, CLL and MM) to SARS-Cov2 vaccine responses and the effects of a third booster. They generally demonstrate that patients with a lack of response to the initial 2 vaccines resulted in lack of response to the booster. A major confounding element of the study pointed in previous reviews involved the tremendous heterogeneity of the cancers (including both indolent and aggressive NHL which is surprising), treatments and extremely limited sample size (43 evaluable but not all had T cell responses done) all of which preclude the ability to generate real conclusions. The differences in responses between CLL and NHL vs MM is significant and thus lowers power as the later studies with normal donor response excluded them. As the study progresses the sample size gets smaller and smaller making even broad conclusions difficult given all the variables impacting the readouts. Increasing sample size of a particular cancer type is needed, especially due to the various treatment regimens particular for each type of cancer, age, disease burden/status all of which must be carefully controlled for as each variable affects statistical power and ability to draw definitive conclusions as to what affects 3rd dose responses. There are simply too many variables and these are very different cancers and treatments to make conclusions from on overall immune effects to any vaccine or challenge.

A prior review recommended showing paired data from each patient and that should definitely be accommodated given the complexity of the responses as well as variability.

Responses to Reviewer comments as of december 16, 2021:

Reviewer #4 (Remarks to the Author):

The authors have addressed the reviewer concerns. The work is timely and will be of interest to the general community.

- **No response.**

Reviewer #5 (Remarks to the Author):

This revised study attempts to assess the humoral and cell-mediated responses of different cancer patients (NHL, CLL and MM) to SARS-Cov2 vaccine responses and the effects of a third booster. They generally demonstrate that patients with a lack of response to the initial 2 vaccines resulted in lack of response to the booster. A major confounding element of the study pointed in previous reviews involved the tremendous heterogeneity of the cancers (including both indolent and aggressive NHL which is surprising), treatments and extremely limited sample size (43 evaluable but not all had T cell responses done) all of which preclude the ability to generate real conclusions. The differences in responses between CLL and NHL vs MM is significant and thus lowers power as the later studies with normal donor response excluded them. As the study progresses the sample size gets smaller and smaller making even broad conclusions difficult given all the variables impacting the readouts.

Increasing sample size of a particular cancer type is needed, especially due to the various treatment regimens particular for each type of cancer, age, disease burden/status all of which must be carefully controlled for as each variable affects statistical power and ability to draw definitive conclusions as to what affects 3rd dose responses. There are simply too many variables and these are very different cancers and treatments to make conclusions from on overall immune effects to any vaccine or challenge.

A prior review recommended showing paired data from each patient and that should definitely be accommodated given the complexity of the responses as well as variability.

- **We added an age matched data set (shown in Supplementary figure 3) confirming that cellular immune response in lymphoma patients after dose 3 as measured by interferon secretion is comparable to immune response in healthy donors after dose 2. This finding strengthens our previous conclusion of a major benefit of the booster dose in immunocompromised patients.**